Native biodiversity collapse in the
eastern Mediterranean. *Proc. R. Soc. B* **288**:
20202469.

ecology, palaeontology

biodiversity collapse, Mediterranean Sea,
Mollusca, Lessepsian invasion, novel ecosystem

**Author for correspondence:**
Paolo G. Albano
e-mail: pgalbano@gmail.com

Electronic supplementary material is available
online at https://doi.org/10.6084/m9.figshare.
c.5253623.

# Native biodiversity collapse in the eastern Mediterranean

Paolo G. Albano[1], Jan Steger[1], Marija Bošnjak[1,2], Beata Dunne[1],
Zara Guifarro[1], Elina Turapova[1], Quan Hua[3], Darrell S. Kaufman[4], Gil Rilov[5]
and Martin Zuschin[1]

[1]Department of Palaeontology, University of Vienna, Althanstrasse 14, 1090 Vienna, Austria
[2]Croatian Natural History Museum, Demetrova 1, Zagreb, Croatia
[3]Australian Nuclear Science and Technology Organisation, Kirrawee DC, New South Wales 2232, Australia
[4]School of Earth and Sustainability, Northern Arizona University, Flagstaff, AZ 86011, USA
[5]National Institute of Oceanography, Israel Oceanographic and Limnological Research (IOLR),
Haifa 3108001, Israel

PGA, 0000-0001-9876-1024; JS, 0000-0001-7021-810X; MB, 0000-0002-1851-1031;
BD, 0000-0002-0732-680X; ZG, 0000-0002-6245-0475; ET, 0000-0001-6942-4352;
QH, 0000-0003-0179-8539; DSK, 0000-0002-7572-1414; GR, 0000-0002-1334-4887;
MZ, 0000-0002-5235-0198

Global warming causes the poleward shift of the trailing edges of marine
ectotherm species distributions. In the semi-enclosed Mediterranean Sea, con-
tinental masses and oceanographic barriers do not allow natural connectivity
with thermophilic species pools: as trailing edges retreat, a net diversity loss
occurs. We quantify this loss on the Israeli shelf, among the warmest areas in
the Mediterranean, by comparing current native molluscan richness with the
historical one obtained from surficial death assemblages. We recorded only
12% and 5% of historically present native species on shallow subtidal soft
and hard substrates, respectively. This is the largest climate-driven regional-
scale diversity loss in the oceans documented to date. By contrast, assem-
blages in the intertidal, more tolerant to climatic extremes, and in the cooler
mesophotic zone show approximately 50% of the historical native richness.
Importantly, approximately 60% of the recorded shallow subtidal native
species do not reach reproductive size, making the shallow shelf a
demographic sink. We predict that, as climate warms, this native biodiversity
collapse will intensify and expand geographically, counteracted only by Indo-
Pacific species entering from the Suez Canal. These assemblages, shaped by
climate warming and biological invasions, give rise to a 'novel ecosystem'
whose restoration to historical baselines is not achievable.

## 1. Introduction

The unprecedented speed of global warming recorded in the last few decades
and the projections for the near future are an increasing threat to marine biodi-
versity [1,2]. One of the most evident consequences of warming is changes in
species distributions, especially of ectotherms that more fully occupy the
extent of latitudes within their thermal tolerance limits [3,4]. Ranges are expected
to expand at the leading (poleward, cold) edge and contract at the trailing (equa-
torward, warm) edge [5]. Along continental margins, such changes in species
distributions cause ecosystem reconfigurations that can be as drastic as a tran-
sition from kelp forest to persistent seaweed turfs, as shown for Western
Australia [6].

A particularly critical situation occurs when such contractions happen in semi-
enclosed basins like the Mediterranean Sea where land masses and oceanographic
barriers constrain the arrival of southern species from contiguous biogeographical
provinces as would have occurred along open continental margins. In this

scenario, a net loss of species richness—a 'biotic attrition' [7]—is expected to occur, leaving taxonomically and functionally depleted biota behind.

The Israeli shelf is one of the warmest areas in the Mediterranean Sea and has experienced a temperature increase of approximately 3°C in the 1980–2013 period, reaching today a summer surface temperature of 32°C [8,9]. It is almost 4000 km away from the Gibraltar Strait that connects the Mediterranean Sea to the Atlantic Ocean, the origin of the Mediterranean biota after the Messinian crisis [10]. Moreover, the northwest African region from Gibraltar south to 20°N latitude (approx. 2000 km of the coastline) has a summer maximum temperature of just 22.3°C [11]. This solid thermic barrier isolates the Mediterranean Sea from the geographically adjacent West African tropical species pool, which repeatedly served as donor of the thermophilic biota that entered the basin during the warmest interglacials of the Pleistocene [12]. This biotic isolation sets the scene for the largest marine biotic attrition ever recorded under a warming climate. In 1869, however, an artificial connection, the Suez Canal, overcame biogeographical constraints and put the basin in direct contact with the tropical Red Sea species pool. Hundreds of species have flooded the basin, the so-called Lessepsian invasion [13,14].

We here quantify historic and current native and non-indigenous species richness along the approximately 200-km-long Israeli shelf from the intertidal to mesophotic depths. We test the hypothesis that a massive collapse of native molluscs has occurred following the extreme warming of the last few decades, while non-indigenous species thrive. We focus on the phylum Mollusca whose taxonomic and functional diversity makes it a good descriptor of benthic assemblages [15–17] and whose durable shells enable the reconstruction of historical species richness from death assemblages, overcoming the lack of directly observed baseline data. Additionally, we measured body size to determine if native and non-indigenous species reach reproductive size and thus form stable or ephemeral populations. The consequences of our results for biodiversity conservation are discussed in the face of accelerating global warming [18].

## 2. Material and methods

### (a) Study area and current and historical datasets

We collected 109 benthos samples at 16 stations along the entire Mediterranean Israeli shelf in the localities and with the devices listed in electronic supplementary material, table S1. In the intertidal, we manually scraped 1 m$^2$ quadrats per sample. In the rocky subtidal down to 25 m, we used suction sampling on 1 m$^2$ quadrats per sample, while in the rocky mesophotic (92 m) we used a rock dredge [19]. In the soft subtidal (10–40 m) and mesophotic (77–83 m) zones, we collected with a van Veen grab. With the exception of the mesophotic, samples were collected in both spring and autumn to cover intra-annual variation. We sieved samples with a 0.5 mm mesh to retain small species, fixed them in ethanol and picked living individuals. Nudibranchs and other shell-less molluscs were not considered. Body size was measured with a ZEISS SteREO Discovery V. 20 microscope and the associated software. We used a calliper for large species (>2 cm). We measured only species with at least 10 living individuals to have a minimum significant sample size for analyses. Only 2 (5%) of the species in the mesophotic samples satisfied this requirement, and this habitat was thus excluded from measurements. In the case of very abundant species (several hundred/thousands of individuals), we randomly selected 30 specimens per station. Species with not fully settled taxonomy (e.g. Chamidae and Ostreidae) or morphology that does not enable unambiguous body size measurement (e.g. Vermetidae) were excluded from measurements. The maximum size for native species and Mediterranean populations of non-indigenous ones was obtained from the literature.

Due to the absence of datasets that precede the major impacts in the basin, we reconstructed the historical species richness from death assemblages, the taxonomically identifiable molluscan remains encountered in the seabed [20]. Due to their slow degradation in the sea, death assemblages act as archives that accumulate information on species, ecological and functional composition over decades to millennia and ultimately enable reconstructing baselines at any spatial scale [21]. Death assemblages were extracted from samples after drying. Before picking the empty shells, the sediment was split, in order to have subsamples with approximately 1000 shells per station. Pelagic, freshwater and terrestrial species were not considered. The abundance of bivalves and polyplacophorans was divided by 2 and 8 (the number of their skeletal parts), respectively, to obtain comparable abundances to living individuals [22]. Due to the particular substrate, intertidal samples did not contain a death assemblage. In this case, we compared the living assemblage with a checklist based on the literature and experts' advice (electronic supplementary material, table S2). Identification was conducted at the species level. Our work is based on approximately 62 000 specimens representing 371 species. The dataset and the literature sources are available in the electronic supplementary material.

At each subtidal sampling station, we quantified the age distribution of shells comprising the death assemblages by radiocarbon dating 9–15 valves. We focused on three native bivalve species that were still extant in the study area. Shells were dated by accelerator mass spectrometry (AMS), using powdered carbonate targets [23,24]. The 149 obtained radiocarbon ages were converted to calendar ages. We report all ages in calendar years before the year of sample collection. The dating and calibrating procedures are described in detail in the electronic supplementary material.

### (b) Data analysis

We quantified assemblage species richness with the coverage-based estimator developed by Chao & Jost [25] using the iNEXT package [26]. This estimator obeys the replication principle and thus behaves intuitively in ratio comparisons. Species diversity loss was quantified by computing the ratio between the current (living assemblages) and the historical (death assemblages) richness at equal coverage (completeness), using the observed richness of the assemblage with greatest coverage and the rarefied richness of the sample with the lowest one. Compared with traditional diversity estimates at fixed sample size, coverage-standardized richness ratios more faithfully represent the true diversity relationship of any two communities [25].

We tested the statistical significance of these ratios by computing bootstrap-derived distributions. For each station and habitat type, we merged abundances of living and death assemblages into a single dataset and resampled it to obtain a living and a death assemblage with the same original sample size (100 iterations). Under the null hypothesis that current and historic richness did not differ (ratio = 100%), we estimated $p$-values by computing the number of bootstrapped values smaller than the observed value with a percentile approach [27].

The difference between the maximum size of individuals in the literature and the maximum size measured in our samples was divided by the literature maximum size to standardize for variation in body size among species, and then box-plotted. We compared the medians of native and non-indigenous species with the Wilcoxon test. We obtained monthly seawater temperature from the Global Ocean Sea Physical Analysis and

Forecasting Product of the European Union Copernicus Marine Service (GLOBAL_ANALYSIS_FORECAST_PHY_001_024 at https://marine.copernicus.eu/). The dataset has a spatial resolution of 1/12° (approx. 8 km) and 50 depth levels. Sea surface and mesophotic temperature were recorded at 0.5 and 92 m depth, respectively. The shapes and medians of annual temperature distribution (averaged over 2016–2019) were compared with the Kolmogorov–Smirnov and Wilcoxon tests, respectively. All analyses were conducted in the R statistical environment [28].

# 3. Results

## (a) Quantification of diversity loss

Current native molluscan richness in shallow subtidal habitats does not exceed 26% of the historical richness at individual stations ($p < 0.01$ at all stations; figure 1$a$). By pooling stations at the habitat scale, these values are 12% and 5% (both $p < 0.01$) on soft and hard substrates, respectively (table 1). In the intertidal, current native richness ranges between 30 and 67% of its historical values and is 50% at the habitat scale. In the mesophotic zone, native richness ranges between 23 and 122% of the historical one, with 60% of the samples showing a ratio not statistically distinguishable from 100% ($p < 0.01$). The current richness in the mesophotic is probably an underestimation because it is based on samples collected in a single season.

By contrast, the non-indigenous component of the assemblages shows a good match between current and historical richness: the ratio ranges between 25 and 82% (52% at the habitat level) for soft substrates and between 58 and 147% (91% at the habitat level) for hard substrates. In half of the stations, this ratio is statistically indistinguishable from 100%. At all stations, this ratio is higher than that of the native species. In the intertidal, the ratio ranges between 33 and 100% and is 100% at the habitat level. The mesophotic samples did not contain a significant non-indigenous component to run the analysis.

In the subtidal, non-indigenous species grew to considerably larger relative size than native species (Wilcoxon $W = 102$, $p = 0.001$ and $W = 38$, $p = 0.01$ on soft and hard substrates, respectively; figure 1$b$). Here, on both types of the substrate, approximately 60% of the native species attained a maximum size smaller than half the maximum size reported in the literature, whereas non-indigenous species had just 16% and 19% of species smaller than half the maximum size on the soft and hard subtidal substrates, respectively. By contrast, in the intertidal, all native species grew larger than half the literature maximum size; the size of the two non-indigenous species reached 8% and 74% of the literature values.

The death assemblages had very diverse ages (table 1). On southern soft substrates, their median age spanned between 125 and 1461 years, demonstrating that they capture pre-Lessepsian conditions. Northern soft and hard substrates showed much younger median ages spanning between 24 and 56 years. Mesophotic death assemblages had a median age of 941 and 23 years on soft and hard substrates, respectively.

# 4. Discussion

## (a) Biodiversity collapse and its causes

The native molluscan biodiversity on the shallow Mediterranean Israeli shelf has collapsed. Although our sampling design included stations across the whole 200-km-long coastline (covering different sedimentological and oceanographic conditions) and a diverse array of habitats, effective sampling methods, a very fine mesh size and multiple seasons, we did not record 88% and 95% of historically present native species on shallow subtidal soft and hard substrates, respectively. This is the largest regional-scale diversity loss in the oceans recorded so far [29], and extends the recent observations of species declines to a whole phylum, over a regional scale and a 0 to 90 m depth gradient [9,30,31]. Additionally, approximately 60% of the detected native species in the shallow subtidal did not reach half the maximum adult size, a proxy of size at first reproduction (see review in the electronic supplementary material, chapter 1.4). Accordingly, most of them form ephemeral populations of juvenile individuals with no or little reproduction potential, probably sourced by larvae coming from different sectors of the basin and/or deeper waters. The shallow Israeli shelf has become a 'black hole sink' for native molluscs, where there is immigration but no back-migration to source populations [32]. These results fit the hypothesis that increasing seawater temperatures renders the warmest parts of the Mediterranean unsuitable for native species, which are mostly of temperate to boreal affinity [10], as recently shown for fishes [33].

The intertidal and mesophotic assemblages did not show such a marked biodiversity loss, due to their different resistance and exposure, respectively, to warming, although the population of the ecologically important intertidal reef-building gastropod *Dendropoma anguliferum* collapsed to near regional extinction [9]. Due to periodic exposure to air or direct sun irradiation, intertidal organisms have adapted to withstand a broader temperature range than recorded in the surrounding water and air [34]. The limpet *Patella caerulea*, one of the most abundant species in our samples, showed an Arrhenius breakpoint temperature (above which cardiac activity drops off dramatically) of approximately 36°C in Sicily, after being collected in winter with a mean seawater temperature of 16.4°C [35]. The periwinkle *Melarhaphe neritoides*, another common species in our samples, experienced heat coma at 38°C and death at 46.3°C in Wales, UK, after having been collected in summer with a seawater temperature of approximately 15°C [36]. These limits may be higher in the warmer parts of the distributional range [37]. By contrast, even shallow subtidal species show lower temperature limits than phylogenetically closely related intertidal ones: on the Mediterranean coast of France, the subtidal cockle *Acanthocardia tuberculata* showed a median lethal temperature after 48 h ($LT_{50}$) of 28.6–30.8°C, significantly lower than the 32.7–34.6°C of the intertidal *Cerastoderma glaucum* collected in coastal lagoons [38]. The subtidal *Donax semistriatus* showed an $LT_{50}$ of 28.9–30.9°C, significantly lower than the 31–33.1°C of *Donax trunculus* from the same lagoons [39]. Interestingly, both of Ansell's papers showed that juveniles have a greater thermal tolerance than adults. Mesophotic ecosystems experience considerably lower temperatures than the shallow subtidal (figure 2). At these depths, seasonal fluctuations have low amplitude, and despite current temperature increases, species still live well within the thermal tolerance limits of native Mediterranean assemblages.

Potential alternative causes of the massive biodiversity loss documented here may be the competition with non-indigenous species, pollution and disease-driven mass

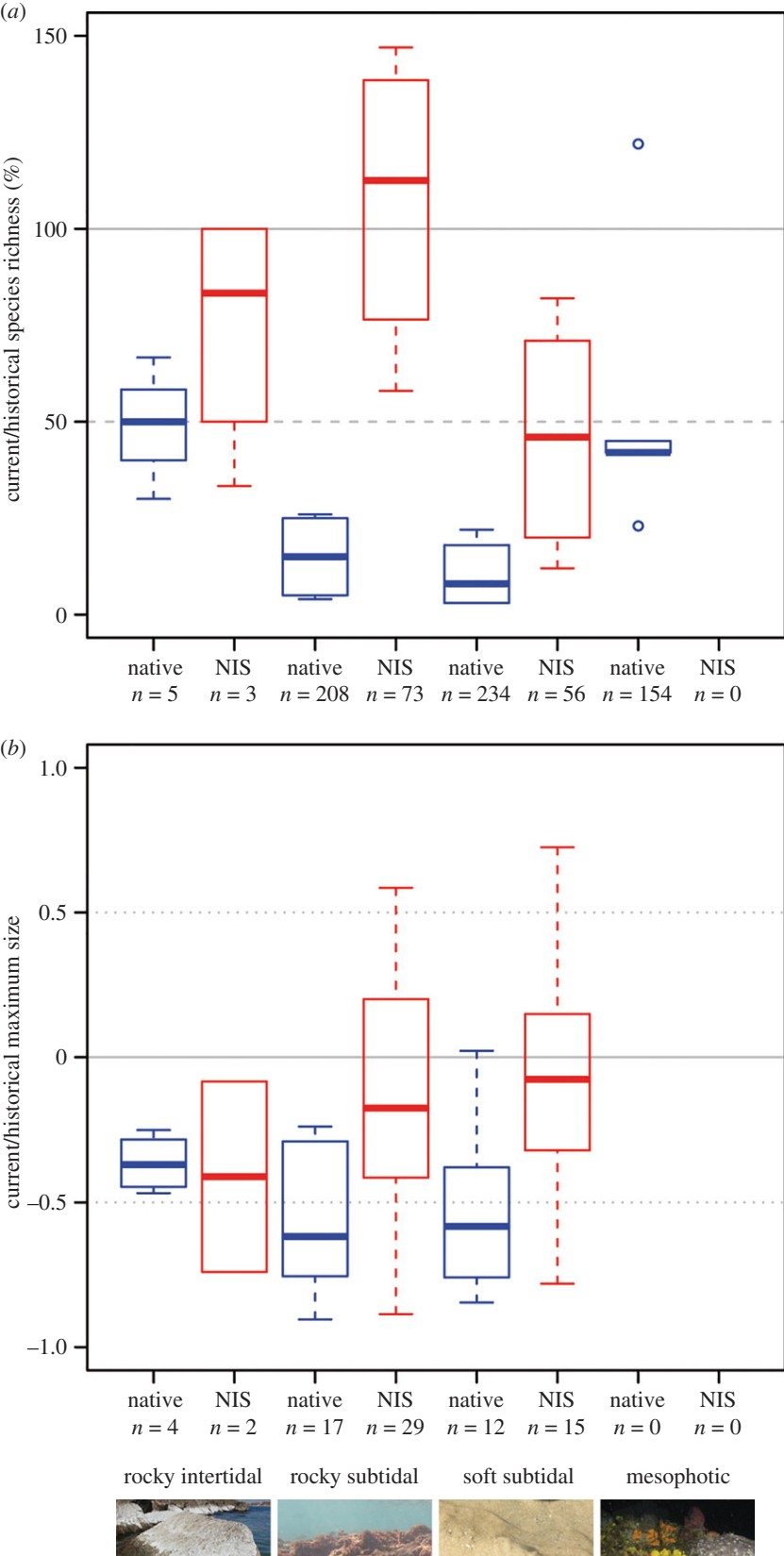

**Figure 1.** (*a*) Distribution of ratios between current and historical richness of the native (blue) and non-indigenous (red) molluscan components on the Mediterranean Israeli shelf (*n*: observed number of species). The mesophotic samples do not have a significant non-indigenous species (NIS) component. Current native species richness is markedly low in the shallow subtidal, and non-indigenous richness always higher than native richness in the same habitat. (*b*) Difference between literature and sample maximum size of the native (blue) and non-indigenous molluscan components (red) on the Mediterranean Israeli shelf (*n*: number of measured species). Negative values mean that the maximum size we recorded is smaller than that in the literature, the opposite for positive values. In contrast to the intertidal, where the six species here analysed constitute 75% of the diversity, the mesophotic samples had only 2 (5%) species with sufficient sample size and were excluded from the analysis. The figure shows that in the shallow subtidal, approximately 60% of the native species do not reach half the maximum size, a proxy for size at first reproduction. By contrast, all intertidal native species were larger than this threshold. Additionally, most NIS were also larger than the size at first reproduction, marking a distinct reproductive potential compared with native species. (Online version in colour.)

**Table 1.** Ratio between current and historical richness on the Mediterranean Israeli shelf, its significance value (bootstrap, 100 iterations; italics indicate significant values, $p < 0.05$) and the median age and 95% age range of death assemblages (DA). The intertidal samples are compared against a checklist compiled from the literature (and thus no significance value could be computed and no age data are available), while the subtidal and mesophotic samples are compared against the corresponding death assemblage at the same coverage (see Materials and Methods). The mesophotic samples come from a single season (TG80: autumn; others: summer); thus the values shown are underestimations; non-indigenous species (NIS) these samples do not have a significant non-indigenous species (NIS) component.

**intertidal**

|  | station scale |  |  |  | habitat scale |  | regional scale (shallow) |
|---|---|---|---|---|---|---|---|
|  | Ashqelon | Tel Aviv | Netanya | Nahariya | mean | s.d. |  |
| native | 0.50 | 0.50 | 0.67 | 0.30 | 0.49 | 0.15 | 0.50 |
| NIS | 0.67 | 1.00 | 1.00 | 0.33 | 0.75 | 0.32 | 1.00 |

**shallow soft (10–40 m)**

|  | station scale |  |  |  |  |  | habitat scale |  | regional scale (shallow) |
|---|---|---|---|---|---|---|---|---|---|
|  | Ashqelon −10 m | Ashqelon −20 m | Ashqelon −30 m | Ashqelon −40 m | Atlit −10 m | Atlit −30 m | mean | s.d. |  |
| median DA age (years) | 763 | 769 | 1461 | 125 | 24 | 53 |  |  |  |
| DA 95% age range (years) | 54–1021 | 28–4733 | 28–5061 | 2–567 | 0–473 | 0–1449 |  |  |  |
| native | 0.22 | 0.18 | 0.03 | 0.07 | 0.03 | 0.09 | 0.10 | 0.08 | 0.12 |
|  | $p = 0$ | $p = 0$ | $p = 0$ | $p = 0$ | $p = 0$ | $p = 0$ |  |  | $p = 0$ |
| NIS | 0.25 | 0.67 | 0.2 | 0.12 | 0.71 | 0.82 | 0.46 | 0.30 | 0.52 |
|  | $p = 0$ | $p = 0$ | $p = 0$ | $p = 0$ | $p = 0.43$ | $p = 0.36$ |  |  | $p = 0$ |

**shallow hard (12–25 m)**

|  | station scale |  |  |  | habitat scale |  | regional scale (shallow) |
|---|---|---|---|---|---|---|---|
|  | Ashqelon −12 m | Ashqelon −25 m | Achziv −11 m | Achziv −20 m | mean | s.d. |  |
| median DA age (years) | 56 | 26 | 55 | 33 |  |  |  |
| DA 95% age range (years) | 1–56 | 1–2912 | 2–165 | 2–58 |  |  |  |
| native | 0.04 | 0.06 | 0.26 | 0.24 | 0.15 | 0.12 | 0.05 |
|  | $p = 0$ | $p = 0$ | $p = 0$ | $p = 0$ |  |  | $p = 0$ |
| NIS | 0.95 | 0.58 | 1.47 | 1.3 | 1.08 | 0.39 | 0.91 |
|  | $p = 0.39$ | $p = 0.01$ | $p = 0.99$ | $p = 0.94$ |  |  | $p = 0.24$ |

**mesophotic (80–90 m)**

|  | station scale |  |  |  |  | habitat scale |  |
|---|---|---|---|---|---|---|---|
|  | RC16 (hard) | RC66 (hard) | CH2 (hard) | other (hard) | TG80 (soft) | mean | s.d. |
| median DA age (years) | 23 | NA | NA | NA | 941 |  |  |
| DA 95% age range (years) | 1–1677 | NA | NA | NA | 271–3864 |  |  |
| native | 0.23 | 1.22 | 0.42 | 0.42 | 0.45 | 0.55 | 0.39 |
|  | $p = 0$ | $p = 0.72$ | $p = 0.12$ | $p = 0.04$ | $p = 0.47$ |  |  |

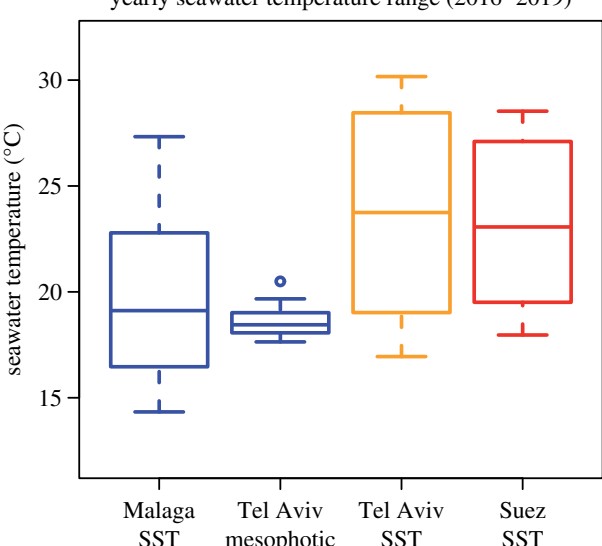

**Figure 2.** Yearly seawater temperature range (averaged over 2016–2019) in the Western (Malaga) and Eastern (Tel Aviv) Mediterranean, and in the Gulf of Suez, northern Red Sea. The surface temperature in Tel Aviv is indistinguishable from that in the Gulf of Suez (Kolmogorov–Smirnov $D = 0.25$, $p = 0.869$; Wilcoxon $W = 68$, $p = 0.843$) and much higher than in the mesophotic zone (K–S $D = 0.67$, $p = 0.008$; $W = 30$, $p = 0.015$). The latter is indistinguishable from the median temperature in Malaga, Western Mediterranean (K–S $D = 0.42$, $p = 0.256$; $W = 66$, $p = 0.755$), although characterized by a much more restricted seasonal amplitude. (Online version in colour.)

mortalities. Competitive exclusion of native by invasive species has been claimed to be strongly impacting biodiversity, and there are indeed multiple examples of competitive exclusion of natives by invaders in terrestrial (e.g. [40,41]) and marine ecosystems (e.g. [42,43]), but species traits and environmental context will eventually determine the actual relationships among native and non-indigenous species, and whether exclusions indeed occur [44–48]. The hundreds of Red Sea species that have established populations on the Israeli shelf [49] have been similarly claimed to have caused the extirpation of some native species (e.g. [50,51]). However, recent studies showed that successful non-indigenous fishes preferentially occupy different functional niches than native ones, providing so far little support for active competitive exclusion of functionally similar species [52–55]. Additionally, the preliminary results of an ongoing functional trait study by our team of the same mollusc assemblages analysed here suggest little potential for active resource competition between native and non-indigenous assemblage components in the shallow subtidal [56]. Conversely, competition, in combination with warming, may have assisted native population collapses of other taxonomic groups in the region, such as sea urchins [57].

Pollution effects on populations of benthic assemblages span from changes in structure to demise and local extinction [58]. Although discharges into the environment (mostly sewage sludge) occur at multiple sites along the Israeli coastline, Haifa Bay is the single pollution hotspot due to heavy metal contamination, still present due to contaminated groundwater even 20 years after the cessation of discharges [59]. Because of this potential bias, Haifa Bay was not included in our sampling design (electronic supplementary material, figure S1). A known cause of molluscan decline is the exposure to endocrine disruptors, such as the organotins tributyltin and

triphenyltin, which cause imposex, the superimposition of male sexual characters onto females, impairing reproduction [60]. Before the ban by the International Maritime Organization in 2008, these compounds were common in anti-fouling paints and occurred predominantly in ports and marinas. Their strongest effects were thus local: sterile neogastropods were recorded up to 1 km from two Mediterranean Israeli marinas [61]. By contrast, the pattern of loss we describe here occurs at broader spatial scales, and our sampling design excluded organotin contamination hotspots (electronic supplementary material, figure S1). Additionally, although imposex has been recorded in 268 gastropod species so far, the majority (213 species, 80%) were neogastropods [62], and although it occurs in bivalves too, it apparently does not affect their recruitment [63]. Our results, however, show massive loss among all molluscan taxa (electronic supplementary material, table S6).

Mass mortality events of marine invertebrates are increasing in frequency [64]. Nonetheless, all the 21 disease-driven events reported by Fey *et al*. affected a single species. Consistently, the iconic pen-shell *Pinna nobilis*, one of the world's largest bivalves, has undergone a mass mortality across the whole Mediterranean Sea since 2016 caused by the parasite *Haplosporidium pinnae* [65] which, however, did not affect the co-generic *Pinna rudis* or other bivalve species. Along the entire Israeli rocky shore, a mass mortality event of the highly abundant non-indigenous mussel *Brachidontes pharaonis* occurred in the summer of 2016 for yet unknown reasons, wiping out the entire population, but not other intertidal molluscs [66]. It is therefore unlikely that diseases could have erased the phylogenetically diverse molluscan assemblages here studied. Additionally, if pollution and diseases played a role in the native species collapse, it is unclear why they did not affect non-indigenous species. Finally, fishing does not affect molluscan assemblages because shelled molluscs are not harvested on the Israeli shelf.

A potential confounding factor is that shells are preserved in surficial marine sediments for decades to millennia, depending on individual species' shell durability and local sedimentation rates [67]. Consequently, death assemblages may contain also species that disappeared before modern human pressures, inflating the baseline historical richness and the perceived magnitude of biodiversity loss. In such a case, biodiversity loss would positively correlate with the median age of death assemblages. Our radiocarbon dating results do not support this hypothesis. On shallow soft substrates, where the median age spanned between 24 and 1461 years (table 1), the magnitude of diversity loss did not correlate with the median age (Spearman $r = 0.17$, $p = 0.74$). Additionally, the hard substrates had even younger median ages (between only 26 and 56 years), but their diversity loss is greater than on soft substrates. Such low median ages imply that the assemblages were considerably richer just a few decades ago and that the biodiversity collapse has occurred in very recent times, as also suggested by Rilov [9].

## (b) Perspectives for the future Mediterranean Sea

The shallow subtidal Israeli shelf has experienced a directional shift from assemblages composed of Mediterranean species to assemblages dominated by tropical non-indigenous ones, to the degree that they are unrecognizable by an observer who is familiar with Mediterranean biota. Assemblage restructuring (massive loss of native species and replacement

by non-indigenous ones) similar to what we show here for molluscs was also observed for soft bottom fish assemblages and rocky reef communities in the same region [30,50]. The easternmost Mediterranean Sea hosts the warmest sectors of the basin along with some endemic species and clades, e.g. in vermetid gastropods [68] and macroalgae [69]. Such entities may be driven to global extinction as the environmental conditions in the basin continue to change at a faster pace than adaptation. We argue that a similar loss of native biodiversity may be under way also in other parts of the eastern Mediterranean that are less thoroughly monitored, and the projected increasing sea temperatures may cause its geographical spread to the western and northern Mediterranean Sea [70,71].

At the same time, the disappearance of native biodiversity paves the way for an even larger-scale biological invasion from the Red Sea. Tropical species are successful invaders in temperate areas if they manage to survive winters and acquire sufficient resources during the warmer periods [72]. Seawater warming is facilitating invaders on both aspects: on the one hand, temperatures on the Mediterranean Israeli shelf are now not dissimilar from those in the Gulf of Suez, the closest donor area (Kolmogorov–Smirnov $D = 0.25$, $p = 0.869$; Wilcoxon $W = 68$, $p = 0.843$; figure 2); on the other hand, the disappearance of native species may increase the resources available to Red Sea invaders. Together, both factors create 'niche opportunities' for non-indigenous species while decreasing the biotic resistance of recipient assemblages against invasions [73,74]. From this perspective, even though the Lessepsian invasion was caused by the construction of the Suez Canal and its subsequent enlargements, its current magnitude may be a consequence of warming.

The eastern Mediterranean shallow subtidal ecosystem is rapidly becoming a 'novel ecosystem' *sensu* Hobbs *et al.* [75] because of both considerable abiotic (due to climate warming) and biotic (due to the Lessepsian invasion) environmental transformations over broad spatial (hundreds of km) and temporal (multiple decades) scales. This new state is probably irreversible. The massive native biodiversity collapse we describe here has occurred under *recent climate change* conditions, but ocean warming will continue even if the global mean surface air temperature can be stabilized at or below 2°C, due to the inertia of the ocean system [76]. Moreover, there are no plans to interrupt the connectivity of the Mediterranean Sea with the Red Sea, and we can expect that non-indigenous species will become even more numerous and dominant, while their eradication is unrealistic. Additionally, our results suggest the need of effective protection of the mesophotic zone (e.g. from fishing and the construction of infrastructure), which still hosts a diverse native assemblage almost devoid of non-indigenous species, that could partly repopulate the shallower shelf, at least with the most eury-bathic species, should conditions ever return to those more typical of the temperate Mediterranean.

Finally, the extreme abiotic conditions and the large number of non-indigenous species suggest that this novel ecosystem may have well crossed thresholds that make the restoration of historical baselines not achievable [77]. Under these circumstances, prioritizing management strategies focusing on ecosystem functions appears a sensible and timely approach [78,79]. Considering that Lessepsian species are likely to be the dominant organisms able to withstand future environmental conditions, understanding their functional traits and ecological roles is of crucial importance for taking informed management decisions. The timeliness of this task cannot be highlighted enough as recent evidence suggests that at least some assemblages dominated by non-indigenous species on the Israeli shallow shelf do not provide the same functions of the previous native assemblages [56,80]. The Eastern Mediterranean is an extreme but revelatory example of the major changes that anthropogenic global stressors are exerting on marine biodiversity.

Ethics. Fieldwork was conducted with permit 41928 of the Israel Nature and Parks Authority.

Data accessibility. Data and code are available as electronic supplementary materials and from the Dryad Digital Repository: https://doi.org/10.5061/dryad.pnvx0k6kk [81].

Authors' contributions. P.G.A. designed the study, acquired and analysed the data and wrote the first draft of the manuscript. J.S., M.B., B.D., Z.G., E.T., G.R. and M.Z. contributed to data acquisition. Q.H. and D.S.K. obtained and calibrated radiocarbon ages. J.S., Q.H., D.S.K., G.R. and M.Z. contributed to the interpretation of results and manuscript writing.

Competing interests. The authors declare no competing interests.

Funding. This project was funded by the Austrian Science Fund (FWF) P28983-B29 (PI: P.G.A.). Shell dating was supported by a grant of the University of Vienna to M.Z.

Acknowledgements. We thank Jonathan Belmaker and Shahar Malamud for their help in organizing and conducting fieldwork, and Dar Golomb and Maura Schonwald for their help in collecting the samples from the mesophotic rocky reefs. Stefan Dullinger offered useful suggestions. Bruno Amati, Michele Azzarone, Cesare Bogi, Patrick Bukenberger, Karolina Czechowska, Davide Di Franco, Ivo Gallmetzer, Justina Givens, Menachem Goren, Alexander Heidenbauer, Anna Hinterplattner, Angelina Ivkić, Henk Mienis, Jan Päßler, Bruno Sabelli and Martina Stockinger helped at various stages. Katherine Whitacre and Jordon Bright prepared radiocarbon samples, which were analysed at the University of California at Irvine Keck AMS Laboratory.

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
