## [Reviewer comments · Proceedings of the Royal Society B: Biological Sciences]

Review History

RSPB-2020-2469.R0 (Original submission)

Review form: Reviewer 1

Recommendation

Accept as is

Scientific importance: Is the manuscript an original and important contribution to its field?

Excellent

General interest: Is the paper of sufficient general interest?

Excellent

Quality of the paper: Is the overall quality of the paper suitable?

Excellent

Is the length of the paper justified?

Yes

Should the paper be seen by a specialist statistical reviewer?

No

Do you have any concerns about statistical analyses in this paper? If so, please specify them explicitly in your report.

No

It is a condition of publication that authors make their supporting data, code and materials available - either as supplementary material or hosted in an external repository. Please rate, if applicable, the supporting data on the following criteria.

Is it accessible?

Yes

Is it clear?

Yes

Is it adequate?

Yes

Do you have any ethical concerns with this paper?

No

Comments to the Author

This study examines the effects of warming sea surface temperatures on shallow marine mollusk assemblages in the Mediterranean using comparisons between live mollusks and accumulating skeletal material, or death assemblages. Death assemblages record historical conditions and are an invaluable source of data for pre-disturbance conditions. The authors document a 88-95% decline in native mollusk species relative to pre-warming assemblages, and an overabundance of small individuals that have not reached reproductive maturity. These findings have broad implications for conservation, which are well described in the manuscript.

The methods and analyses are appropriate, particularly the use of historical assemblages to document biodiversity loss. The authors provide an excellent discussion of alternative hypotheses and draw reasonable conclusions from their results. This is a timely and well-constructed paper providing vital insights into the magnitude of the effects of warming on shallow marine ecosystems. This topic is of broad interest to a wide readership, making PRSB an excellent venue for publication.

While the number of native species shared in the live and dead assemblages was low, I am curious as to why no size comparisons were done for those species in the habitats with robust dead samples? The death assemblage provides an excellent opportunity for sympatric historical size comparisons, and it would be quite interesting to see how the proportion of specimens above the size for sexual maturity differs between the live and dead. I hope the authors have plans to publish data of this nature in a future manuscript.

Review form: Reviewer 2

Recommendation

Accept with minor revision (please list in comments)

Scientific importance: Is the manuscript an original and important contribution to its field?

Good

General interest: Is the paper of sufficient general interest?

Good

Quality of the paper: Is the overall quality of the paper suitable?

Good

Is the length of the paper justified?

Yes

Should the paper be seen by a specialist statistical reviewer?

No

Do you have any concerns about statistical analyses in this paper? If so, please specify them explicitly in your report.

No

It is a condition of publication that authors make their supporting data, code and materials available - either as supplementary material or hosted in an external repository. Please rate, if applicable, the supporting data on the following criteria.

Is it accessible?

Yes

Is it clear?

Yes

Is it adequate?

Yes

Do you have any ethical concerns with this paper?

No

Comments to the Author

The ms 'Native biodiversity collapse in the Eastern Mediterranean' by Albano and co-workers provides first-hand data on the dramatic decline of autochthonous marine communities (using molluscs as a suitable surrogate) in face of increasing warming and alien pressure in the highly-representative Eastern Mediterranean. The AA supply experimental statistically-validated own data, and the outcome of their study is used to predict future ecological scenarios, including the onset of new communities replacing the historical ones.

The ms is well organized and written, and in my view the article is suitable for being considered by PRS B with minor modifications (not substantial) to the text (all reported in the annotated text, attached).

I advise the AA to be at places less assertive in their consequentialness, a case in point being line 33 in the Abstract: 'As climate warms...willl'. A more conservative statement such ' We predict that, as climate warms etc' is preferable.

Decision letter (RSPB-2020-2469.R0)

16-Nov-2020

Dear Dr Albano:

Your manuscript has now been peer reviewed and the reviews have been assessed by an Associate Editor. The reviewers' comments (not including confidential comments to the Editor) and the comments from the Associate Editor are included at the end of this email for your reference. As you will see, the reviewers and the Editors have raised some concerns with your manuscript and we would like to invite you to revise your manuscript to address them.

Research ethics:

Use of animals and field studies:

It is a condition of publication that you make available the data and research materials supporting the results in the article. Please see our Data Sharing Policies (<https://royalsociety.org/journals/authors/author-guidelines/#data>). Datasets should be deposited in an appropriate publicly available repository and details of the associated accession number, link or DOI to the datasets must be included in the Data Accessibility section of the article (<https://royalsociety.org/journals/ethics-policies/data-sharing-mining/>). Reference(s) to datasets should also be included in the reference list of the article with DOIs (where available).

Please submit a copy of your revised paper within three weeks. If we do not hear from you within this time your manuscript will be rejected. If you are unable to meet this deadline please let us know as soon as possible, as we may be able to grant a short extension.

Best wishes,
Dr Daniel Costa
mailto:proceedingsb@royalsociety.org

Associate Editor

Board Member: 1

Comments to Author:

Reviewer 1 was positive about the ms but called for historical vs modern comparisons of size where species occurred at both time intervals. This might provide insight into how the proportion of specimens above the size for sexual maturity differs between the live and dead assemblages. Reviewer 2 was also quite positive but had a number of stylistic comments and edits on the ms itself (see attached pdf). We look forward to seeing your revision along with your response to the reviewer's concerns.

Reviewer(s)' Comments to Author:

Referee: 1

Comments to the Author(s)

This study examines the effects of warming sea surface temperatures on shallow marine mollusk assemblages in the Mediterranean using comparisons between live mollusks and accumulating skeletal material, or death assemblages. Death assemblages record historical conditions and are an invaluable source of data for pre-disturbance conditions. The authors document a 88-95% decline in native mollusk species relative to pre-warming assemblages, and an overabundance of small individuals that have not reached reproductive maturity. These findings have broad implications for conservation, which are well described in the manuscript.

The methods and analyses are appropriate, particularly the use of historical assemblages to document biodiversity loss. The authors provide an excellent discussion of alternative hypotheses and draw reasonable conclusions from their results. This is a timely and well-constructed paper providing vital insights into the magnitude of the effects of warming on shallow marine ecosystems. This topic is of broad interest to a wide readership, making PRSB an excellent venue for publication.

While the number of native species shared in the live and dead assemblages was low, I am curious as to why no size comparisons were done for those species in the habitats with robust dead samples? The death assemblage provides an excellent opportunity for sympatric historical size comparisons, and it would be quite interesting to see how the proportion of specimens above the size for sexual maturity differs between the live and dead. I hope the authors have plans to publish data of this nature in a future manuscript.

Referee: 2

Comments to the Author(s)

The ms 'Native biodiversity collapse in the Eastern Mediterranean' by Albano and co-workers provides first-hand data on the dramatic decline of autochthonous marine communities (using molluscs as a suitable surrogate) in face of increasing warming and alien pressure in the highly-representative Eastern Mediterranean. The AA supply experimental statistically-validated own data, and the outcome of their study is used to predict future ecological scenarios, including the onset of new communities replacing the historical ones.

The ms is well organized and written, and in my view the article is suitable for being considered by PRS B with minor modifications (not substantial) to the text (all reported in the annotated text, attached).

I advise the AA to be at places less assertive in their consequentialness, a case in point being line 33 in the Abstract: 'As climate warms...willl'. A more conservative statement such 'We predict that, as climate warms etc' is preferable.

Author's Response to Decision Letter for (RSPB-2020-2469.R0)

See Appendix A.

Decision letter (RSPB-2020-2469.R1)

07-Dec-2020

Dear Dr Albano

I am pleased to inform you that your manuscript entitled "Native biodiversity collapse in the Eastern Mediterranean" has been accepted for publication in Proceedings B.

Open Access

Paper charges

Sincerely,

Dr Daniel Costa
Editor, Proceedings B
<mailto:proceedingsb@royalsociety.org>

Associate Editor:

Board Member

Comments to Author:

(There are no comments.)

Appendix A

Referee 1

This study examines the effects of warming sea surface temperatures on shallow marine mollusk assemblages in the Mediterranean using comparisons between live mollusks and accumulating skeletal material, or death assemblages. Death assemblages record historical conditions and are an invaluable source of data for pre-disturbance conditions. The authors document a 88-95% decline in native mollusk species relative to pre-warming assemblages, and an overabundance of small individuals that have not reached reproductive maturity. These findings have broad implications for conservation, which are well described in the manuscript.

The methods and analyses are appropriate, particularly the use of historical assemblages to document biodiversity loss. The authors provide an excellent discussion of alternative hypotheses and draw reasonable conclusions from their results. This is a timely and well-constructed paper providing vital insights into the magnitude of the effects of warming on shallow marine ecosystems. This topic is of broad interest to a wide readership, making PRSB an excellent venue for publication.

While the number of native species shared in the live and dead assemblages was low, I am curious as to why no size comparisons were done for those species in the habitats with robust dead samples? The death assemblage provides an excellent opportunity for sympatric historical size comparisons, and it would be quite interesting to see how the proportion of specimens above the size for sexual maturity differs between the live and dead. I hope the authors have plans to publish data of this nature in a future manuscript.

PGA: There are two reasons why we did not compare the size of empty shells in the death assemblages with the maximum size as reported in the literature.

First, death assemblages are not preserved in the intertidal setting. These samples were collected from rocks by scraping and thus constitute only living individuals. The comparison of the size of living assemblages vs the maximum size from the literature enables a coherent methodological framework to address the question on species reproductive potential irrespective of the habitat or the sampling technique.

Second, even though we processed more than 10,000 empty shells, some species found alive were poorly represented in the death assemblage. For example, from the subtidal hard substrates we recovered 23 and 38 native and non-indigenous species, respectively, with more than 10 living individuals; of these, only 10 (44%) and 13 (34%) species, respectively, were represented by at least 10 empty shells. This implies that the comparison of the size of empty shells in the death assemblage with the reproductive size from the literature would have been limited to a small fraction of the diversity only (the referee refers to this problem too). By comparing the size in the living assemblages with the maximum size from the literature, we were able to include 17 (74%) and 29 (76%) of the native and non-indigenous species, respectively, removing only species with uncertain taxonomic status or whose morphology does not enable unambiguous size measurements (as specified in the methods in lines 99-100).

We certainly plan to publish data of this nature in a future manuscript, but such a study would not cover the array of habitats and species here inspected and thus would not enable the broad understanding of the underlying processes we offer in the present manuscript.

Referee 2

The ms 'Native biodiversity collapse in the Eastern Mediterranean' by Albano and co-workers provides first-hand data on the dramatic decline of autochthonous marine

communities (using molluscs as a suitable surrogate) in face of increasing warming and alien pressure in the highly-representative Eastern Mediterranean. The AA supply experimental statistically-validated own data, and the outcome of their study is used to predict future ecological scenarios, including the onset of new communities replacing the historical ones.

The ms is well organized and written, and in my view the article is suitable for being considered by PRS B with minor modifications (not substantial) to the text (all reported in the annotated text, attached).

PGA: I here address each reviewer's comment citing the line numbers in the annotated pdf:

Lines 29-30: suggested words added to the abstract.

Line 60: the distance between Gibraltar and the Israeli shelf has been specified.

Lines 86-89: the terminology to be applied to marine depth zones is indeed not unambiguous in the literature. The reviewer highlighted that, whereas the intertidal zone is defined by geometric limits, the mesophotic is not, being instead defined by light penetration. The reviewer further suggests the use of the terms infralittoral and circalittoral in place of shallow subtidal and mesophotic, respectively. Unfortunately, this suggestion does not fully solve the problem because the boundary between the infralittoral and the circalittoral is also defined by light penetration: the lower limit of the infralittoral is defined as the depth "...compatible with the existence of marine phanerogams or photophilous algae (which have the same needs, as regards light, as the phanerogams)" (Pérès 1967 *The Mediterranean benthos, Oceanography and Marine Biology: an Annual Review* 5: 449-533, which is the English version of the Pérès & Picard's "Nouveau manuel de bionomie benthique de la Mer Méditerranée" published in 1964 where the terms "infralittoral" and "circalittoral" were introduced). Therefore, we prefer to stick to our terminology which is currently more common in the literature and for each depth zone inspected, we specified the depth intervals of our samples in meters.

Line 324: the reviewer suggested to replace "global" with "regional" when we discussed the effect of continuing warming on endemic clades in the Eastern Mediterranean. Our point is the following: we demonstrated that the current environmental conditions brought the native fauna on the Israeli shelf to the brink of regional eradication (at present, we have no evidence of global extinction because none of the species we found dead only had a distribution restricted to the Israeli shelf). With increasing warming, these adverse conditions may spread to the whole Eastern Mediterranean. Because this basin hosts endemic species, the native species eradication at the basin scale would imply their total disappearance from the globe, and thus we wrote of "global extinctions". Highlighting "global" is very important, because the reported extinctions in the marine environment are very few (just 20 species according to the review by Harnik et al 2012 *TREE* 27: 608-617). Warming in a hotspot of endemism like the Mediterranean Sea challenges the view that marine species are less prone to global extinctions.

Line 336: we followed the suggestion by replacing "increase" with "may increase".

Line 336: Oliverio & Taviani 2003 addressed a very important topic: after the last glaciation, the Eastern Mediterranean has had environmental conditions suitable for hosting a tropical fauna. The closest donor would have been tropical West Africa, separated from the Mediterranean by the cold water barrier off Mauritania and Morocco. The Eastern Mediterranean had then a potential (in terms of 'niche opportunities') for tropical species, which could be expressed only after the opening of the Suez Canal, when non-indigenous species of Indo-Pacific origin entered the

basin. We dedicated an entire manuscript to this topic, currently under review, where we precisely demonstrated that these tropical non-indigenous species have different ecological traits than their native counterparts perfectly fitting the predictions of Oliverio & Taviani 2003. We added the citation of Oliverio & Taviani 2003 to our manuscript.

Line 351: we specified that much can be done to protect the mesophotic zone from stressors other than warming and biological invasions. In particular, forbidding destructive fishing techniques and the construction of infrastructure (e.g. oil and gas pipelines).

I advise the AA to be at places less assertive in their consequentialness, a case in point being line 33 in the Abstract: 'As climate warms...willl' . A more conservative statement such ' We predict that, as climate warms etc' is preferable.

PGA: We have modified the sentence as suggested.